# Second-Neighbor Hopping Effects in the Two-Dimensional Attractive Hubbard Model

Rodrigo Alves Fontenele, Nathan Vasconcelos, Natanael Carvalho Costa, Thereza Paiva *
and Raimundo Rocha dos Santos

Instituto de Física, Universidade Federal do Rio de Janeiro, Cx.P. 68.528, Rio de Janeiro 21941-972, Brazil
* Correspondence: tclp@if.ufrj.br

**Abstract:** The emergence of superconductivity (SC) in lattice models, such as the attractive Hubbard one, has renewed interest since the realization of cold-atom experiments. However, reducing the temperature in these experiments is a bottleneck; therefore, investigating how to increase the energy scale for SC is crucial to cold atoms. In view of this, we examine the effects of next-nearest-neighbor hoppings ($t'$) on the pairing properties of the attractive Hubbard model in a square lattice. To this end, we analyze the model through unbiased Quantum Monte Carlo simulations for fixed density $n = 0.87$, and perform finite-size scaling analysis to the thermodynamic limit. As our main result, we notice that the existence of further hopping channels leads to an enhancement of the pairing correlations, which, in turn, increases the ground-state order parameter $\Delta$. Finally, at finite temperatures, for $t'/t \neq 0$, this enhancement of pairing correlations leads to an increase in the critical temperature $T_c$. That is, the fine-tuning of second-neighbor hoppings increases the energy scales for SC, and may be a route by which cold-atom experiments can achieve such a phase and to help us further understand the nature of this phenomenon.

**Keywords:** superconductivity; Hubbard model; quantum Monte Carlo





## 1. Introduction

The attractive Hubbard model (AHM) is a standard phenomenological model describing the transition from a high-temperature metallic or insulating state to a low-temperature superconducting state [1], through the dynamics of fermions moving subject to an on-site interaction, $U < 0$, favoring the formation of local pairs. This attractive interaction can arise when some degrees of freedom—phononic or excitonic—are eliminated [2–4]. Over the years this model has played an important role in elucidating many aspects of superconductivity, such as the existence of relevant pseudogap phenomena in high-temperature cuprate superconductors [5], in addition to its important role in the study of the physical properties of strongly correlated fermionic systems. For instance, a recent extensive Quantum Monte Carlo (QMC) mapping of the square lattice phase diagram, $T_c(\langle n \rangle, U)$, where $\langle n \rangle$ is the band-filling, indicates a maximum $T_c \approx 0.15t/k_B$ ($t$ is the hopping integral and $k_B$ is the Boltzmann constant; we set the ratio $t/k_B$ to unity from this point) of around $\langle n \rangle \approx 0.87$ [6].

The emergence and continuing development of experiments in optical lattices, in which ultracold fermionic atoms are loaded and the interaction amongst them controlled through Feshbach resonances [7–11], enabled the study of the AHM in an unprecedented way. Many properties of the AHM have been measured and analyzed through this setup [12]. Notwithstanding the major experimental advances, the lowest temperatures achieved to date are around 0.45, still about three times the predicted maximum $T_c$ for a square lattice.

Thus, a natural question is whether we can generate a more robust superconducting phase by changing the band structure, perhaps through the introduction of next-nearest-neighbor (NNN) hopping, $t'$. Indeed, by allowing fermions to hop along the diagonals of a square lattice, one provides additional paths for Cooper pairs to move around, thus

decreasing the possibility of pair-breaking scattering events taking place; this simple picture is supported by perturbative approaches [13] and by early QMC simulations for a limited set of parameters [14], both showing an enhancement of superconducting pairing correlations when $t'$ is switched on. Concerning the possible experimental realizations of NNN hopping, we note that a triangular optical lattice was experimentally set up by adding an extra pair of counter-propagating laser beams; see Ref. [15]. Given that the triangular lattice may be thought of as a square lattice with hopping along one of the diagonals, a possible route to generate NNN hopping on a square lattice would be to add yet another pair of counter-propagating laser beams, this time in the other diagonal direction.

In view of this, it is certainly of interest to investigate how far the critical temperature can be increased for values of $|U|$ beyond the weak-coupling approximation.

Here, we report results from Monte Carlo simulations showing that NNN hopping increases the superconducting gap. The layout of the paper is as follows. In Section 2, we discuss the model and highlight the main aspects of DQMC, including the quantities used to probe the physical properties of the system. In Section 3 we present estimates for both the zero-temperature gap and the critical temperature for non-zero second-neighbor hopping, and Section 4 presents our final conclusions.

## 2. Model and Methodology

The attractive Hubbard Hamiltonian with second-neighbor hopping [13,14,16] reads

$$\mathcal{H} = \sum_{\mathbf{i},\mathbf{j},\sigma} t_{\mathbf{i},\mathbf{j}} \left( c^{\dagger}_{\mathbf{i},\sigma} c_{\mathbf{j},\sigma} + H.c. \right) - \mu \sum_{\mathbf{i},\sigma} n_{\mathbf{i},\sigma} - |U| \sum_{\mathbf{i}} \left( n_{\mathbf{i}\uparrow} - 1/2 \right) \left( n_{\mathbf{i}\downarrow} - 1/2 \right) \tag{1}$$

where the sums run over all sites of a square lattice, $H.c.$ stands for the Hermitian conjugate, $c^{\dagger}_{\mathbf{i},\sigma}$ ($c_{\mathbf{i},\sigma}$) are creation (annihilation) operators of electrons on given sites $\mathbf{i}$ with spin $\sigma$, while $n_{\mathbf{i},\sigma} = c^{\dagger}_{\mathbf{i},\sigma} c_{\mathbf{i},\sigma}$ are the number operators in the conventional second quantization formalism. Here, we define the hopping integral as

$$t_{\mathbf{i},\mathbf{j}} = \begin{cases} t, & \text{if } \mathbf{i}, \mathbf{j} \text{ are nearest neighbors (NN)}, \\ t', & \text{if } \mathbf{i}, \mathbf{j} \text{ are next-nearest neighbors (NNN)}, \\ 0, & \text{otherwise}, \end{cases} \tag{2}$$

and $\mu$ is the chemical potential controlling the band filling; the NN hopping integral $t = -1$ sets the energy scale. The last term in (1) corresponds to the local attractive interaction between electrons, with coupling strength $|U|$.

From the outset, one should note that NNN hopping gives rise to subtle effects, such as the breakdown of particle–hole symmetry and the destruction of Fermi surface nesting at half filling. Indeed, Figure 1 shows the non-interacting density of states (DOS) for $t' = 0$ and for $t' = -0.2$, in which case the van Hove singularity corresponds to a filling near our density of interest, $\langle n \rangle = 0.87$: there is, therefore, a huge increase in the DOS, which, by virtue of the BCS argument [1], $T_c \sim W \exp\left(-1/D_0 U\right)$, where $W$ is the band width and $D_0$ is the DOS at the Fermi energy, should cause an increase in the critical temperature. Nonetheless, Figure 1 also shows that $D_0$ can also decrease for some band fillings, so that the net effect in $T_c$ ultimately depends on a delicate balance between $U$, $t'$, and $\langle n \rangle$.

We investigated the finite temperature properties of the AHM by performing determinant quantum Monte Carlo (DQMC) simulations [3,17–20]. The DQMC method is an unbiased numerical approach based on an auxiliary-field decomposition of the interaction, which maps onto a quadratic form of free fermions coupled to bosonic degrees of freedom $\mathcal{S}(i, \tau)$ in both spatial and imaginary time coordinates. This method is based on a separation of the non-commuting parts of the Hamiltonian by means of the Trotter–Suzuki decomposition, i.e.,

$$\mathcal{Z} = \text{Tr}\, e^{-\beta \hat{\mathcal{H}}} = \text{Tr}\left[ \left( e^{-\Delta\tau(\hat{\mathcal{H}}_0 + \hat{\mathcal{H}}_U)} \right)^M \right] \approx \text{Tr}\left[ e^{-\Delta\tau \hat{\mathcal{H}}_0} e^{-\Delta\tau \hat{\mathcal{H}}_U} e^{-\Delta\tau \hat{\mathcal{H}}_0} e^{-\Delta\tau \hat{\mathcal{H}}_U} \cdots \right], \tag{3}$$

where $\hat{\mathcal{H}}_0 = \sum_{\mathbf{i,j},\sigma} t_{\mathbf{i,j}} \left( c^\dagger_{\mathbf{i},\sigma} c_{\mathbf{j},\sigma} + H.c. \right) - \mu \sum_{\mathbf{i},\sigma} n_{\mathbf{i},\sigma}$, and $\hat{\mathcal{H}}_U = -|U| \sum_{\mathbf{i}} (n_{\mathbf{i}\uparrow} - 1/2)(n_{\mathbf{i}\downarrow} - 1/2)$. This decomposition leads to an error proportional to $(\Delta\tau)^2$, which can be systematically reduced as $\Delta\tau \to 0$. Throughout this work, we chose $\Delta\tau \leq 0.1$ (depending on the temperature), which is small enough to lead to systematic errors that are smaller than the statistical ones (from the Monte Carlo sampling). Finally, for the AHM, the discrete Hubbard–Stratonovich transformation dealing with the quartic terms in $\hat{\mathcal{H}}_U$ leads to sign-free simulations [3,18–20].

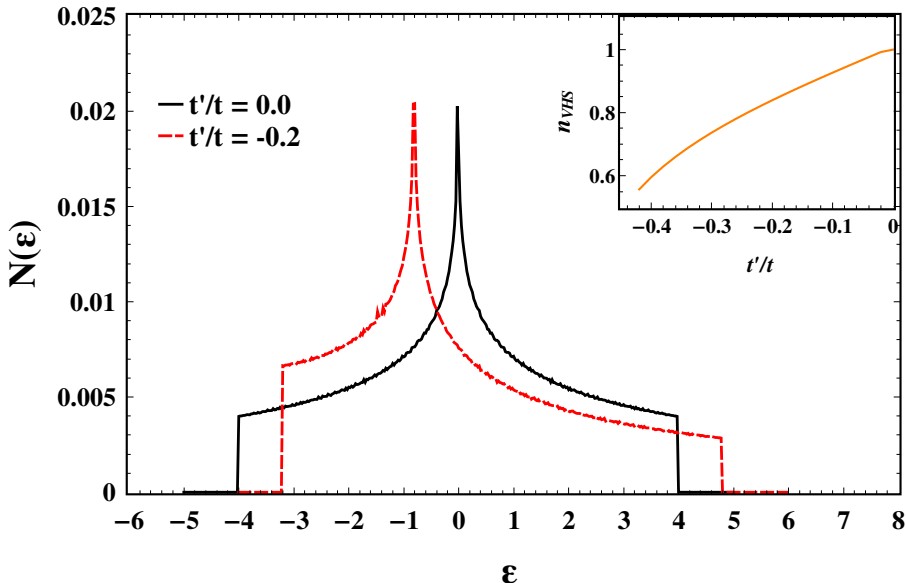

**Figure 1.** (Color online) Density of states $N(\epsilon)$ for $t' = 0$ and $t' = 0.2$. The inset shows the electronic density at which the van Hove singularity occurs, $n_{VHS}$, as a function of the NNN-hopping $t'/t$.

In order to probe the emergence of superconductivity, we analyzed the *s*-wave pair correlation functions,

$$C^\Delta_{\mathbf{ij}} \equiv \frac{1}{2} \langle b^\dagger_{\mathbf{i}} b_{\mathbf{j}} + H.c. \rangle, \tag{4}$$

with $b^\dagger_{\mathbf{i}} \equiv c^\dagger_{\mathbf{i}\uparrow} c^\dagger_{\mathbf{i}\downarrow}$ ($b_{\mathbf{i}} \equiv c_{\mathbf{i}\downarrow} c_{\mathbf{i}\uparrow}$) corresponding to the creation (annihilation) of a pair of electrons at site a given site $\mathbf{i}$. Further, the Fourier transform of $C^\Delta_{\mathbf{ij}}$ at $\mathbf{q} = 0$ defines the *s*-wave pair structure factor,

$$P_s(\mathbf{q} = 0) = \sum_{\mathbf{i,j}} \frac{1}{N} C^\Delta_{\mathbf{ij}} \tag{5}$$

with $N = L \times L$ being the number of sites of the lattice. Finally, we estimate the critical temperature, $T_c$, using the correlation ratio,

$$R_c(L) = 1 - \frac{P_s(\mathbf{q} + \delta\mathbf{q})}{P_s(\mathbf{q})}, \tag{6}$$

where $\mathbf{q} = 0$, and $\delta\mathbf{q} = 2\pi/L$. This quantity is a renormalisation-group invariant at the critical point [21–23]; that is, the crossings of $R_c(L)$ for different system sizes provide estimates for $T_c$. Here, we consider a lattice with linear sizes $L = 14$–$20$, which are large enough to reduce finite size effects.

## 3. Results and Discussion

We start by discussing the subtle case of the system at half electronic filling, $\langle n_{\mathbf{i}} \rangle = 1$. For this filling, and fixing $t' = 0$, we recall that charge-density wave (CDW) and singlet superconductivity (SS) are degenerate; therefore, no long-range order emerges at finite temperatures, for any $|U|$, as a result of the Mermin–Wagner theorem [24]. As one dopes away from half filling, CDW correlations are suppressed and a Kosterlitz–Thouless (KT)

phase transition to a SS phase occurs at finite values $T_c$; see, e.g., Ref. [6] and references therein for discussions. However, when $t' \neq 0$, charge correlations may be frustrated, thus breaking the CDW/SS degeneracy even at $\langle n_{\mathbf{i}} \rangle = 1$. Therefore, the case of half-filling is not representative of the effects brought about by NNN hopping.

Given this, here we investigate how $t'$ affects the superconducting properties away from half filling, namely, at $\langle n_{\mathbf{i}} \rangle \approx 0.87$, which is the optimal doping for $T_c$ in the absence of NNN hoppings [6]. For the non-interacting case, the main effect of second-neighbor hopping is to shift the van Hove singularity in the density of states (DOS), located at half filling, to lower densities [25]; if $t'/t$ is large, then we expect that the critical temperature would be suppressed as a response to the reduction in the DOS. By the same token, one may naively expect (from BCS theory) that, if such a shifted singularity were close to the investigated filling, the critical temperatures would increase. Therefore, in what follows, we fix $t'/t = -0.2$, which leads to an enhancement in the non-interacting DOS at $\langle n_{\mathbf{i}} \rangle \approx 0.87$.

Figure 2 shows the decay of $C_{ij}^{\Delta}$ with the distance for (a) $t'/t = 0$, and (b) $t'/t = -0.2$, at different temperatures, and fixed $|U|/t = 3$. The behaviour is similar in both cases, i.e., fast decay at high temperatures, due to the lack of pair coherence, giving rise to a much slower decay at low temperatures, suggestive of superconducting order in the ground state. Further, a closer look at the data seems to indicate that $C_{ij}^{\Delta}(\mathbf{r}_{ij})$ stabilizes at a slightly larger value for $t'/t = -0.2$ than for $t'/t = 0$. In order to check this, in Figure 3a we show the $s$-wave structure factor, $P_s$, as a function of the inverse temperature, $\beta t$. We see that all data stabilize at low temperatures, at values which increase with the system size, consistently with order in the ground state. In addition, for a given system size, the data for $t' \neq 0$ stabilize at larger values than those for $t' = 0$.

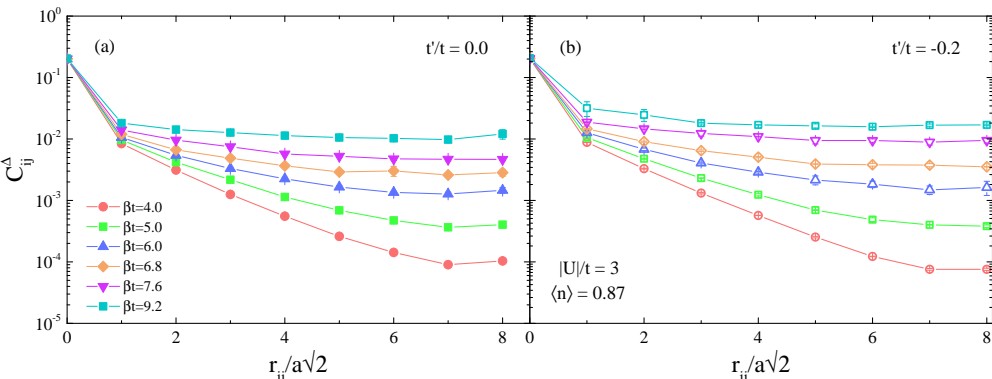

**Figure 2.** (Color online) Log-linear plot of the decay with diagonal distance of the pairing correlation function for different temperatures, at fixed $U/t = -3$, $\langle n \rangle = 0.87$, for (**a**) $t'/t = 0$, and (**b**) $t'/t = -0.2$.

This can be cast into a more quantitative basis by performing a finite-size scaling analysis of $P_s$ through the Huse ansatz [26–28],

$$\frac{P_s}{L^2} = \Delta_0^2 + \frac{b}{L^2}, \quad T \to 0, \tag{7}$$

where $\Delta_0$ is the superconducting gap function at zero temperature, and $b \equiv b(U, t')$ is independent of $L$. Accordingly, the extrapolated (to $\beta t \to \infty$) data of $P_s$ for both $t' = 0$ and $t' \neq 0$ are plotted in Figure 3b as functions of $1/L^2$. The intersections with the vertical axis provide estimates for $\Delta_0^2$, which are larger for $t' \neq 0$ than for $t' = 0$, consistent with a more robust pairing. By invoking the BCS result, $T_c \sim \Delta_0$, we may expect that, for this band filling, $T_c$ is enhanced for $t'/t = -0.2$.

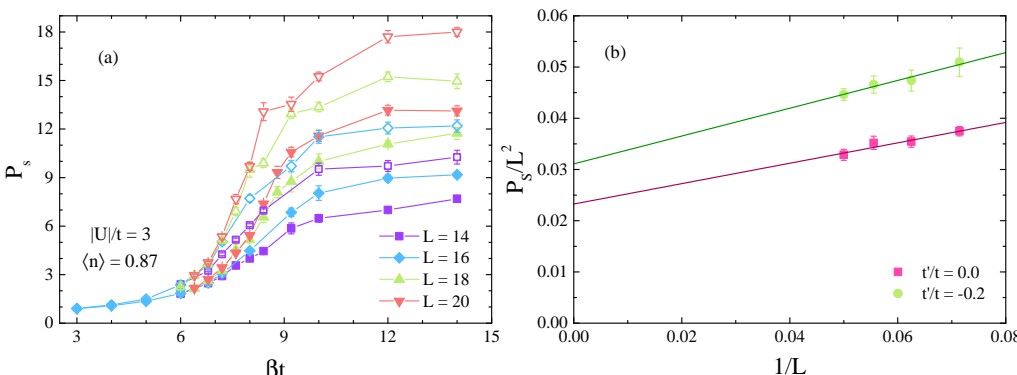

**Figure 3.** (Color online) Panel (**a**) shows the Pair structure factor $P_s$ as a function of inverse temperature, for different linear lattice sizes $L$. Squares are used for $14 \times 14$ lattices, diamonds represent $16 \times 16$ lattices, up triangles for $18 \times 18$ ones, and down triangles for $20 \times 20$ lattices. Panel (**b**) shows the extrapolation with $1/L$ of the saturated values of $P_s/L^2$. In both panels, the full symbols correspond to $t'/t = 0$ results, while the empty symbols to $t'/t = -0.2$ ones.

Let us now estimate $T_c$ through the correlation ratio, $R_c$, Equation (6). Figure 4 shows the temperature dependence of $R_c(L)$ for $t' = 0$ [panel (a)] and for $t'/t = -0.2$ [panel (b)], for three different system sizes. For sufficiently large system sizes, the curves should cross at a single point, $\beta_c$. However, for the sizes considered here, these crossings occur at slightly different values of $\beta$: while, for $t'/t = 0.0$ the intersection occurs at $\beta t \approx 8.0 \pm 0.2$ ($T_c/t \approx 0.125 \pm 0.003$), for $t'/t = -0.2$, we estimate $\beta t \approx 7.5 \pm 0.2$ ($T_c/t \approx 0.133 \pm 0.004$). That is, $T_c$ for the NNN hopping case is slightly enhanced with respect to the NN hopping.

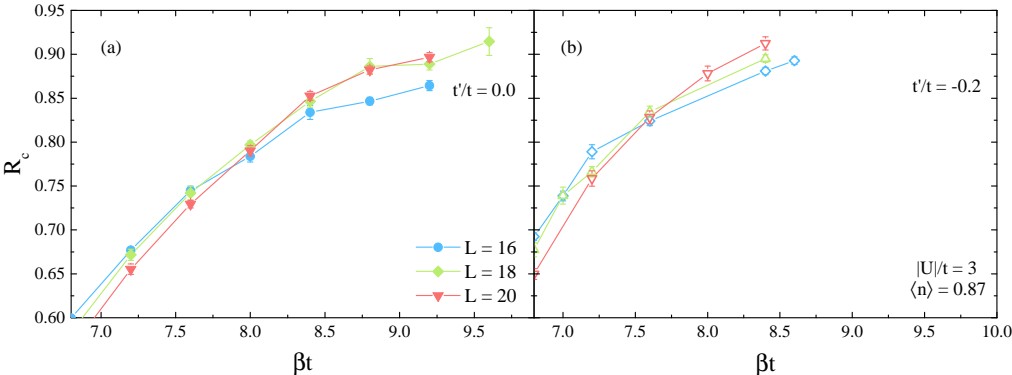

**Figure 4.** (Color online) The correlation ratio $R_c$ as a function of the inverse of temperature, for different lattice sizes, $\langle n \rangle \approx 0.87$, and $U/t = -3$. In panel (**a**) we have $t'/t = 0.0$, whereas in (**b**) $t'/t = -0.2$.

## 4. Conclusions

We have examined the effects of NNN hopping on some pairing properties of the attractive Hubbard model, for fixed $U/t = -3$, and electronic density $\langle n \rangle \approx 0.87$. We analyzed the pairing correlation functions by determinant quantum Monte Carlo simulations for system sizes up to $N = 20 \times 20$ and different temperatures. Our main result is that we noticed an enhancement of pairing correlations when $t'/t \neq 0$, which leads to a larger order parameter in the ground state and a higher critical temperature. Physically, this may be attributed to an increase in the number of hopping channels, which amounts to an electron pair being less likely to be trapped as a result of the Pauli principle. We expect our findings will stimulate further studies of the second-neighbor hopping as a way to increase the critical temperature for superfluidity to the point of making phase transition accessible in cold-atom experiments.

**Author Contributions:** Conceptualization, T.P., R.R.d.S. and N.C.C.; formal analysis, R.A.F., N.V., N.C.C., T.P. and R.R.d.S.; writing—original draft preparation, and writing—review and editing, R.A.F., N.C.C., T.P. and R.R.d.S. All authors have read and agreed to the published version of the manuscript.

**Funding:** This research was funded by Fundação Carlos Chagas Filho de Amparo à Pesquisa do Estado do Rio de Janeiro grant number E-26/200.959/2022 (T.P.) and grant number E-26/200.258/2023 - SEI-260003/000623/2023 (N.C.C.); National Council for Scientific and Technological Development grant number 313065/2021-7 (N.C.C.), 305871/2017-0 (R.R.d.S.), 140985/2020-4 (R.A.F.), 403130/2021-2 (T.P.) and 308335/2019-8 (T.P.); and Instituto Nacional de Ciência e Tecnologia de Informação Quântica, grant number 465469/2014-0 (T.P. and R.R.d.S.)

**Informed Consent Statement:** Not applicable.

**Data Availability Statement:** Data available on request.

**Acknowledgments:** The authors are grateful to the Brazilian Agencies Conselho Nacional de Desenvolvimento Científico e Tecnológico (CNPq), Coordenação de Aperfeiçoamento de Pessoal de Ensino Superior (CAPES), Fundação Carlos Chagas de Apoio à Pesquisa do Estado do Rio de Janeiro (FAPERJ), and Instituto Nacional de Ciência e Tecnologia de Informação Quântica (INCT-IQ) for funding this project.

**Conflicts of Interest:** The authors declare no conflict of interest. The funders had no role in the design of the study; in the collection, analyses, or interpretation of data; in the writing of the manuscript, or in the decision to publish the results.

## Abbreviations

The following abbreviations are used in this manuscript:

| | |
|---|---|
| AHM | Attractive Hubbard model |
| CDW | Charge-density wave |
| DOS | density of states |
| NN | nearest neighbor |
| NNN | next-nearest neighbor |
| QMC | Quantum Monte Carlo |
| DQMC | Determinant Quantum Monte Carlo |

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
