# Peer review of "Second-Neighbor Hopping Effects in the Two-Dimensional Attractive Hubbard Model"

_condensedmatter, doi:10.3390/condmat8010011_

Round 1

Reviewer 1 Report

In this manuscript, the authors have investigated the effects of next-nearest-neighbor hopping on the pairing properties in the attractive Hubbard model.
The result is interesting but I would like to ask several questions before the decision.

1. The dimension is not mentioned explicitly (I guess that they consider the two-dimensional
system). Accordingly, I would like to ask the authors about the relationship between this
work and the experimental paper J. K. Chin, et al Nature 443, 961-964 (2006);
https://www.nature.com/articles/nature05224 Is the main difference the dimensionality?

2. It would be better to comment on the repulsive Hubbard model, which is also relevant to the recent cold atom experiment (e.g., A. Mazurenko, et al Nature 545, 462–466 (2017);  https://www.nature.com/articles/nature22362).
Naively, can we expect the enhanced Tc even in the repulsive case due to the next-nearest-neighbor hopping?

3. Eq. (5) seems to already take q=0, but Eq. (6) indicates q-dependence of P_s. Probably the phase factor exp(iqr) would be dropped in Eq. (5).

4. The definition of the critical temperature given by Eq. (6) seems to be similar to the Thouless criterion where the pair correlation function diverges at zero energy and momentum (e.g., for 3D critical temperature, Nozieres and Schmitt-Rink (1985); https://link.springer.com/article/10.1007/BF00683774 ).
In this regard, it would be different from that for the Berezinskii-Kosterlitz-Thouless transition temperature in 2D.
It would be better if the authors could address this point.

5. Do the authors have any idea to control the next-nearest-neighbor hopping t’ in cold atom
experiments? Is it possible to realize relatively shallow optical lattice or superlattices?

6. Regarding the enhancement of Tc by the next-nearest-neighbor hopping, the density of states (DOS) around the Fermi level might be useful to understand physics.
Namely, if the DOS around the Fermi level increases due to t', T_c may also increase by following the BCS argument. Is it possible to show the DOS (non-interacting one would be sufficient)?

Author Response

Please find comments and answer in the pdf file attached.

Reviewer 2 Report

In the presented work entitled „ Second-neighbor hopping effects in the attractive Hubbard model” by R. A. Fontele et al., the Authors synthesize analyze the attractive Hubbard model in context of the cold atom experiments. In particular they test how the higher-order hopping terms help in matching the theoretical estimates with the experimental predictions.

The presented manuscript is mostly written in a clear and well organized manner, while the analysis seems to be free of any major errors.  However, there are few points that must be clarified before my final decision. In particular:

1.     My first comment is related to the motivation behind conducted research. In the cold atom experiments, the reported temperatures are up to three times higher than their theoretical counterparts. It means that the theoretical model does not describe well the superconducting phases of interest. The Authors propose to introduce higher order hopping terms to match predictions of the theoretical model with the experiment. However, what is the rationale behind this approach? For example, the high-order hopping terms are known to “fine-tune” the band structure by breaking certain crystal symmetries like the symmetry around the Fermi level. Should we expect some similar effects here, or maybe the problem is much more complex? Please explain at least briefly (please do not refer to other studies only).

2.     In relation to the above, I suggest the Authors to add more “physics” into their discussion. For example, the Introduction refer to the hopping parameters multiple times but what these terms exactly mean for the analysis/problem. This is to say, please explain what is the physical meaning of the discussed parameters and what are the reasons to analyze them in the first place. Such explanation will be of great value to the less experienced readers. However, at the same time we will be looking at more “physical” discussion than just a fitting model discussion.

3.     Why the Authors decide to consider calculations only up to the second-nearest-neighbor hopping terms? Why they are not considering higher terms? What is the physical reasoning behind such choices?

4.     Are the presented calculations self-consistent? How the number operators are estimated and what is the exact procedure? If they are simply assumed, do the Authors expect that their values are actually correctly corresponding to the other parameters values (like the hoppings) in the model? In other words, do the Authors actually work with the ground states?

5.     The Authors choose the square lattice model for their considerations. Please explain why?

6.     Please discuss explicitly the novelty of the conducted research and obtained results in terms of other studies.

*end of report*

Author Response

Please find comments and answers in the pdf file attached.

Round 2

Reviewer 1 Report

I appreciate the authors' effort to address all the comments.

While I am basically satisfied with the revised manuscript and the reply,

I am still afraid that the definition of P_s(q) in Eq. (5) is insufficient.

If I am correct, one may wonder how the q-dependence of P_s(q) in Eq. (6) is implemented because P_s in Eq. (5) is independent of q.  

If P_s in Eq. (5) and P_s(q) in Eq. (6) are different, the definition of P_s(q) in Eq. (6) would be clarified explicitly. 

Author Response

We have added q=0 in equation 5 to clarify its q dependece.